# An Illusion of Barriers to Gene Flow in Suburban Coyotes (*Canis latrans*): Spatial and Temporal Population Structure across a Fragmented Landscape in Southern California

Savanah Bird [1,2,*], Javier D. Monzón [3], Wallace M. Meyer III [2] and Jonathan E. Moore [2]

[1] Biology Department, University of Oregon, Eugene, OR 97403, USA
[2] Biology Department, Pomona College, Claremont, CA 91711, USA; wallace.meyer@pomona.edu (W.M.M.III); jon.moore@pomona.edu (J.E.M.)
[3] Natural Science Division, Pepperdine University, Malibu, CA 90263, USA; javier.monzon@pepperdine.edu
[*] Correspondence: savanahb@uoregon.edu

**Abstract:** Carnivores with large home ranges are especially vulnerable to habitat fragmentation. As coyotes (*Canis latrans*) are often found living in highly modified landscapes, it is unclear how urban and suburban development impact gene flow between their populations. This study evaluated gene flow among coyotes inhabiting California sage scrub fragments within the highly developed Pomona Valley, California. We genotyped microsatellites from scat samples collected from four study sites to examine population structure between coyotes separated by a major freeway, coyotes separated by suburban development, and finally, coyotes in contiguous, natural habitat sites over 15 months. Though coyotes from all four sites were genetically distinct, near-complete turnover of individuals in sites and examination of temporal genetic structure and relatedness within one site indicated the movement of family groups through natural fragments over time. Thus, we argue that solely examining spatial genetic structure may create the illusion of genetic barriers among coyote populations where they may not exist, and that incorporating temporal components of genetic variation is critical to understanding gene flow across space and time in highly mobile animals. Understanding how to better study and manage coyotes, an apex predator, is key to the conservation of the endangered California sage scrub ecosystem.

**Keywords:** habitat fragmentation; gene flow; population structure; southern California





## 1. Introduction

Habitat fragmentation can reduce dispersal and lead to the reproductive isolation of small populations, a phenomenon particularly detrimental to the preservation of biodiversity [1]. Small, isolated populations are more prone to suffer from a loss of genetic diversity, which can increase the prevalence of deleterious genetic traits [2–4], and, in extreme cases, may result in local extinction [5]. Urbanization especially contributes to isolation through loss, fragmentation, and degradation of habitat and is thus capable of rapidly impacting a variety of vertebrate species [6,7].

In low-elevation areas of southern California, natural habitats remain in the foothills and as small fragments or 'islands' surrounded by a 'sea' of high-density urban and suburban development in the valleys. Additionally, an intricate network of multi-lane freeways intersects these developed areas. Fragments are typically composed of one of three vegetation types: (1) endangered native shrublands dominated by drought-deciduous sage scrub or evergreen chaparral; (2) non-native grasslands dominated by various European grass species; or (3) non-native forblands typically dominated by invasive mustard species (e.g., *Brassica* spp. and/or *Hirschfeldia incana*) [8]. These fragments also host a variety of endemic and, in some cases, vulnerable vertebrate species [9].

In southern California, coyotes (*Canis latrans*) are often found in small natural habitat fragments as well as the developed matrix that surrounds those fragments. While an omnivore, the coyote often acts as the de facto apex predator in these fragments, as other apex predators, such as mountain lions (*Puma concolor*), occur in very low density and avoid urban development [10]. Coyotes thus play important ecological roles, such as regulating meso-predator populations, particularly non-native cats [11,12], which is key to preserving bird and reptile diversity [13–15]. Because their presence is critical to preserving regional biotas and ecosystem function, it is imperative to understand how coyote populations are maintained within habitat fragments.

The coyote provides a highly relevant case study for examining the effects of habitat fragmentation and urban development on dispersal and population genetics of carnivores in urban and suburban landscapes. Coyotes are able to persist in urban and suburban environments, sometimes subsisting on anthropogenic food sources [16]. Some coyotes spend most of their lives exclusively in developed areas [17,18]. Consequently, coyotes are behaviorally plastic carnivores typically thought to be unaffected by urbanization. However, if coyotes are impacted by urbanization, it provides a clear indication that these landscape modifications impact more sensitive wildlife.

Previous studies have examined connectivity among coyote populations in urban settings of southern California. Tigas et al. [19] found that radio-collared coyotes in a suburban landscape regularly crossed major roads and maintained home ranges that spanned more than one naturally vegetated fragment. Riley et al. [20] found that a small proportion of sampled coyotes (4.5%) crossed the Ventura Freeway, a large multi-lane freeway in southern California, in a 7-year period. While both studies suggest that major roads and freeways are not impermeable, they still represent substantial barriers to movement. Coyote populations on either side of the Ventura Freeway and a secondary road were genetically differentiated, and coyotes that crossed the freeway rarely reproduced [20]. Additionally, coyote home range boundaries generally followed freeway borders, resulting in a pile-up of territories along them, suggesting that the freeways create artificial territory boundaries that influence reproductive behavior and thus are barriers to gene flow. More recently, Adducci et al. [21] found that coyotes in urban areas of Greater Los Angeles are genetically distinct and less diverse than those in the surrounding mountains, suggesting that urbanization reduces gene flow and accelerates the loss of genetic diversity.

Our study aims to further explore the extent to which a major freeway and the suburban matrix function as barriers to gene flow in coyote populations across the Pomona Valley of southern California. In contrast to previous studies, our study examines population dynamics across both space and time, in an ecological context that contains fragments isolated from contiguous habitats by suburban development alone, and by suburban development and a freeway. In the taxonomy of threats to biodiversity implemented by the International Union for Conservation of Nature, coyotes in our study area are impacted by threat categories 1.1 (housing and urban areas) and 4.1 (roads and railroads) [22]. Because coyotes are commonly observed in suburban areas, we hypothesized that suburban development is not a barrier to gene flow; we predicted that coyotes in sites isolated by suburban development only are not genetically distinct from those in contiguous habitats nearby. Conversely, we predicted that sites separated by the freeway would be genetically distinct, as freeways contribute to habitat fragmentation and reduce dispersal of many species, including coyotes [6,20,23,24].

## 2. Materials and Methods

### 2.1. Study Sites

We collected coyote scat from sites near the border of Los Angeles and San Bernardino Counties, and south of the Angeles National Forest (Figure 1). We sampled coyotes from four study sites to assess population structure in areas separated by suburban development only and areas separated by a portion of California State Route 210 (SR 210), an eight-lane heavily trafficked freeway, constructed in the 1990s and opened in 2002.

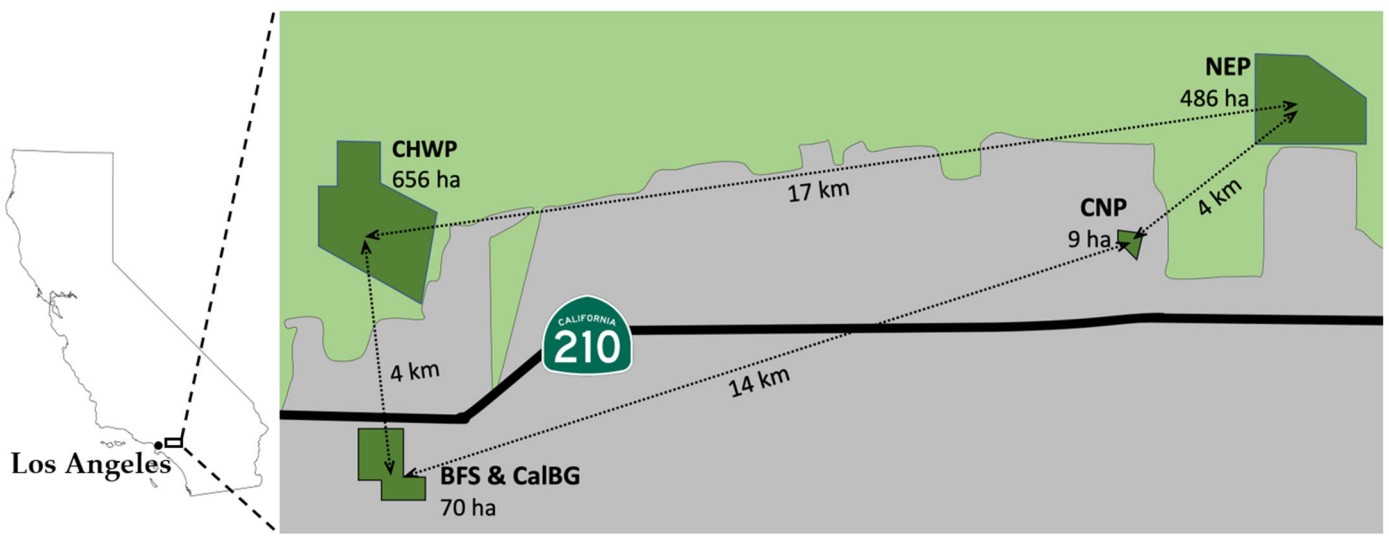

**Figure 1.** Map of study sites (in dark green) for coyote scat collection. Areas dominated by either native or invasive species are green. Suburbia and mining pits are gray. BFS = Bernard Field Station, CalBG = California Botanic Garden, CHWP = Claremont Hills Wilderness Park, NEP = North Etiwanda Preserve, CNP = Chaffey Nature Preserve.

Our southern-most site consisted of both the Bernard Field Station (BFS) and California Botanic Garden (CalBG) in the city of Claremont, two adjacent parcels of land totaling 70 hectares separated by a fence through which coyotes move freely (observations W.M.M.). The vegetation at the BFS consists of California sage scrub and nonnative grasses, and the vegetation of CalBG consists of a wide variety of California native plant species. We collectively refer to these two parcels as the BFS. Four km north of BFS and on the other side of SR 210 is Claremont Hills Wilderness Park (CHWP), which is connected to the Angeles National Forest and consists of mostly non-native grasses and some oak forest. The North Etiwanda Preserve (NEP), 17 km east of CHWP, consists of California sage scrub and is also connected to the Angeles National Forest. Thus, both NEP and CHWP are connected by contiguous natural habitat. Our final site, Chaffey Nature Preserve (CNP), is a small (9 hectare) California sage scrub fragment separated from NEP by suburban development, but no major freeway.

### 2.2. Scat Collection

At each of the four study sites, we opportunistically collected coyote scat along human-made trails approximately once every two months from September 2017 to November 2018. In the field, we distinguished coyote scat from other large carnivore scat by its heterogeneous texture, tapered ends, and presence of seeds or hair. We only collected scat that visually appeared fresh (approximately 2 days old), leaving scat which was excessively dry or sun-bleached. We preserved all scat samples in a −20°C freezer in individual sealed plastic bags [25].

### 2.3. Laboratory Methods

We extracted DNA from each scat using the QIAamp DNA Stool Mini Kit, following kit instructions for large-volume stool samples (Qiagen Inc., Valencia, CA, USA). To confirm the species identification of each scat as coyote, we performed polymerase chain reaction (PCR) testing on the cytochrome b gene [26] using the following mix in a 20-μL reaction volume: 2 μL of template DNA, 0.5 μM of each ScatID primer (Table S1), 1X Standard Taq Reaction Buffer (New England BioLabs Inc., Ipswich, MA, USA), 1.25 mM of MgCl$_2$, 1.6 mg/mL of BSA, 1.5 units of Taq DNA polymerase (New England BioLabs Inc., Ipswich, MA, USA), and 0.2 mM of dNTPS. We amplified cytochrome b using the following thermal cycling program: initial denaturation at 95 °C for 10 min; 55 cycles of 95 °C for 30 s, 52 °C

for 30 sec, and 72 °C for 40 s; and final extension of 72 °C for 3 min. We followed this by standard Taq$\alpha$1 restriction digests of 5 μL of PCR product in a 20-μL reaction volume using 20 units of enzyme (New England BioLabs Inc., Ipswich, MA, USA). We subsequently analyzed restriction digests by agarose gel electrophoresis to confirm coyote identity, using a coyote positive control and a domestic dog (*Canis lupus familiaris*) negative control [27].

On scat samples that were positively identified as coyote, we performed two replicates of PCR on seven tetranucleotide microsatellite loci developed for the domestic dog: FH2001, FH2010, FH2062, FH2096, FH2137, FH2140 [28], and PEZ19 [29]. We used the following protocol: 25-μL reaction volume containing 2 μL of template DNA, 0.5 μM of each primer (Table S1), 1X Standard Taq Reaction Buffer, 1 mM of MgCl$_2$, 1.6 mg/mL of BSA, 1.25 units of Taq DNA polymerase, and 0.2 mM of dNTPS. We included a negative control of PCR components with no template with each set of PCRs. We amplified the microsatellites using the following program: initial denaturation at 95 °C for 5 min; 45 cycles of 95 °C for 30 s; 58 °C for 30 s (62 °C for the FH2096 primer set); and 72 °C for 1 min; and final extension at 72 °C for 15 min. We ran all PCR in Mastercycler thermocyclers (Eppendorf, Hamburg, Germany) in 96-well plates. To prevent DNA contamination, we used separate workspaces and equipment for pre-PCR and post-PCR procedures. We sent PCR products to Cornell University's Biotechnology Resource Center for fragment analysis using ABI 3730xl DNA Analyzers.

To determine whether additional replicates were needed, we assessed congruence between the first and second PCR replicates. Of 197 samples confirmed as coyote from restriction enzyme digestion, 186 samples amplified microsatellites. Of 1,302 possible genotypes (186 coyote scat samples genotyped at 7 loci), we found only 28 cases (2%) of allelic dropout and 18 cases (1%) of otherwise incongruous genotypes between the two replicates. The incongruous genotypes were likely due to spurious peaks in fragment analysis graphs. Because we found strong agreement between the first and second PCR replicates, we determined that only two replicates would be necessary for reliable genotypes.

### 2.4. Genetic Analysis

We distinguished individual coyotes from our samples using the R package allelematch [30], as this program can handle low-quality genotyping data from non-invasively collected samples. In the following analyses, we assumed that samples with the same 5-locus genotype were the same individual (allowing up to 2 allele mis-matches), as probability of identity analysis showed that 5 loci were sufficient to discern siblings with a ~1% probability of match. For each study site we calculated the observed heterozygosity, expected heterozygosity, and average allelic richness using GenAlEx 6.5 [31] and assessed deviations from Hardy–Weinberg equilibrium across all loci using Fisher's exact test in GENEPOP [32]. We also used Fisher's exact test to assess Hardy–Weinberg disequilibrium for each locus within each site. In the Hardy–Weinberg tests, we based significance on a Bonferroni correction for multiple comparisons [33].

To evaluate population differentiation, we calculated global and pairwise fixation indices ($F_{ST}$) among the coyotes from each site and tested the significance via 9999 permutations using GenAlEx 6.5 [31]. To evaluate population structure across our sites, we performed genetic assignment tests using STRUCTURE [34], a Bayesian algorithm that groups the entire set of genotyped individuals into K populations that are genetically similar without a priori population assignments. To find the best-fitting value of K, we utilized a correlated allele frequency model with a burn-in of 50,000 followed by 500,000 MCMC reps, allowing admixture. We evaluated a range of K values from 1 to 10, with 10 replicates of each, then selected the highest likelihood value of K using the Evanno method via Structure Harvester [35]. We aligned and averaged the five replicate cluster membership coefficient matrices using CLUMPP [36]. To evaluate relatedness among individual coyotes, we calculated pairwise relatedness coefficients using the Lynch–Ritland relatedness estimator [37]. We estimated mean pairwise relatedness within sites using 9999 random permutations and 10,000 bootstraps in GenAlEx [31].

Because we found evidence of population turnover, we decided to conduct post-hoc analyses examining temporal genetic structure within sites. However, site BFS was the only site with sufficient sample sizes in both years to permit this analysis. We calculated $F_{ST}$ and the significance of differentiation between BFS coyotes sampled in 2017 and BFS coyotes sampled in 2018 using GenAlEx 6.5 [31]. Coyotes sampled in both 2017 and 2018 were included in both year groups for the $F_{ST}$ calculation.

## 3. Results

We collected 220 scat samples over a 15-month sampling period, with 197 confirmed coyote genetic samples. We excluded 50 genetic samples because of low amplification rates (>2 missing loci), and 79 samples with genotypes matching at 5 or more loci. In total, we genotyped 68 individual coyotes across the four sites (29 from BFS, 11 CHWP, 13 CNP, and 15 NEP). No individual was sampled from more than one site. Some individuals had incomplete genotypes, which we treated as missing data in downstream analyses; however, the overall genotyping success rate in the final dataset was 89.3%.

All seven microsatellites were highly polymorphic, and while none deviated from Hardy–Weinberg equilibrium overall, two exhibited heterozygote deficiency in CHWP (Table 1). Across all loci, only the BFS coyotes exhibited heterozygote deficiency (Table 2). Overall, CNP, the smallest fragment, exhibited the lowest genetic diversity, with an observed heterozygosity of 0.47 and an average allelic richness of 3.1. The other sites had relatively higher average allelic richness (>4), and NEP had the highest observed heterozygosity of 0.78 (Table 2). Despite disparities in sample sizes, average allelic richness did not increase with the number of coyotes sampled per site (Table 2).

**Table 1.** Observed heterozygosity, allelic richness, and Hardy–Weinberg deviation by locus.

| | $H_o$ (A)(H-W) * | | | | | | |
|---|---|---|---|---|---|---|---|
| **Site** | **FH2001** | **FH2010** | **FH2062** | **FH2096** | **FH2137** | **FH2140** | **PEZ19** |
| BFS | 0.59 (4) $p = 0.005$ | 0.53 (5) $p = 0.139$ | 0.48 (4) $p = 0.776$ | 0.62 (3) $p = 0.521$ | 0.76 (7) $p = 0.118$ | 0.82 (4) $p = 0.002$ | 0.57 (5) $p = 0.021$ |
| CHWP | 1.00 (5) $p = 0.237$ | 0.70 (4) $p = 0.250$ | 0.57 (6) $p = 0.161$ | 0.91 (6) $p = 0.776$ | 0.33 (3) $p = 0.225$ | **0.18 (5) (−)** **$p = 0.001$** | **0.27 (5) (−)** **$p < 0.0001$** |
| CNP | 0.46 (3) $p = 1.000$ | 0.30 (3) $p = 1.000$ | 0.62 (4) $p = 0.260$ | 0.62 (3) $p = 0.850$ | 0.55 (4) $p = 0.596$ | 0.54 (3) $p = 1.000$ | 0.23 (2) $p = 1.000$ |
| NEP | 0.40 (3) $p = 0.589$ | 0.92 (5) $p = 0.837$ | 0.93 (4) $p = 0.429$ | 0.73 (4) $p = 0.706$ | 0.85 (4) $p = 0.141$ | 0.87 (7) $p = 0.200$ | 0.79 (5) $p = 0.181$ |
| All ** | 0.61 (7) $p = 0.074$ | 0.61 (6) $p = 0.047$ | 0.65 (6) $p = 0.160$ | 0.72 (6) $p = 0.069$ | 0.62 (7) $p = 0.0190$ | 0.60 (11) $p = 0.181$ | 0.46 (6) $p = 0.008$ |

* Observed heterozygosity ($H_o$), allelic richness (A), and deviation from Hardy–Weinberg equilibrium (H-W), where (−) indicates a heterozygote deficit. Bolded *p*-Value indicates significant deviation from Hardy–Weinberg equilibrium following Bonferroni correction. ** Average observed heterozygosity and total allelic richness per locus across all sites.

**Table 2.** Measures of genetic variation in four sampling sites averaged across seven loci.

| **Site** | **N** | **$H_o$** | **$H_e$** | **A** | **H-W** |
|---|---|---|---|---|---|
| BFS | 29 | $0.62 \pm 0.05$ | $0.63 \pm 0.05$ | $4.6 \pm 0.48$ | **(−) $p = 0.006$** |
| CHWP | 11 | $0.57 \pm 0.12$ | $0.65 \pm 0.05$ | $4.9 \pm 0.40$ | $p = 0.062$ |
| CNP | 13 | $0.47 \pm 0.06$ | $0.46 \pm 0.07$ | $3.1 \pm 0.26$ | $p = 0.212$ |
| NEP | 15 | $0.78 \pm 0.07$ | $0.69 \pm 0.05$ | $4.6 \pm 0.48$ | $p = 0.014$ |
| All | 68 | $0.61 \pm 0.04$ | $0.61 \pm 0.03$ | $4.3 \pm 0.24$ | $p = 0.0528$ |

Observed heterozygosity ($H_o$), expected heterozygosity ($H_e$), and allelic richness (A) averaged across all loci, and deviation from Hardy–Weinberg equilibrium (H-W). Standard error ($\pm$) is shown for $H_o$, $H_e$, and A. Bolded *p*-Value indicates significant deviation from Hardy–Weinberg equilibrium following Bonferroni correction.

Overall, coyotes in our four sites were significantly differentiated (global $F_{ST} = 0.196$, $p < 0.001$). Additionally, all six pairwise comparisons produced $F_{ST}$ values significantly

higher than zero (Table 3). The BFS and NEP coyotes were the most similar ($F_{ST}$ = 0.085), and CHWP and CNP coyotes were the least similar ($F_{ST}$ = 0.180). STRUCTURE assignment results showed that five genetic clusters best fit the data (Figure 2 and Figure S1). Each site had an associated genetic cluster with the exception of BFS, which had two clusters. Of 68 coyotes, 62 (91%) had 50% or greater assignment to the genetic cluster associated with their site (Figure 2).

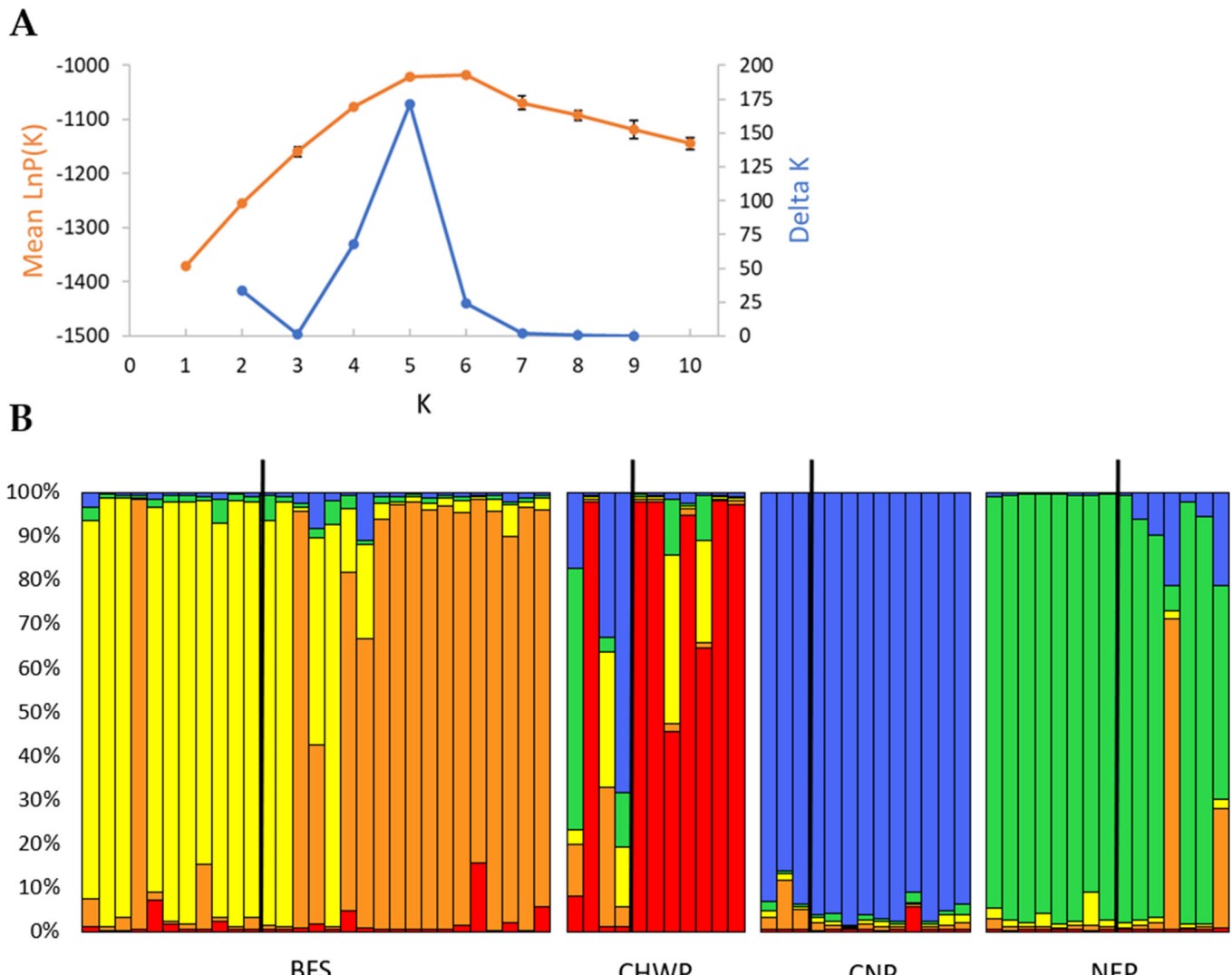

**Figure 2.** (**A**) Evanno plot from STRUCTURE analysis showing that K = 5 is the best−fit model of our data. Orange line shows mean log-likelihood probability of each K value and blue line shows delta K values. (**B**) Genetic assignments from STRUCTURE analysis with five genetic clusters. Plot is aligned and averaged across 10 replicates. The fraction of color for each individual coyote (columns) represents the probability of assignment to that genetic cluster. Samples are sorted temporally by sample date within each sampling location. Black bars within sites separate sampling years 2017 and 2018. For coyotes sampled more than once, we retained the earliest sample with the most complete genotype.

All of our sites showed a turnover of individuals throughout the study period. In the BFS, only 4 of the 29 individual coyotes were sampled in both 2017 and 2018. In both our contiguous sites, NEP and CHWP, just a single individual was sampled in both years. In CNP, no individuals from 2017 were sampled again in 2018. The BFS coyote population itself showed genetic structure (Figure 2), with genetic clusters following a temporal pattern: yellow-cluster coyotes appeared more in 2017 and orange-cluster coyotes

appeared more in 2018. The BFS coyotes in each sample year represented genetically distinct groups ($F_{ST}$ = 0.050, *p* = 0.012). Although other sites appear to also have a small shift in genetic structure between years (Figure 2), low sample sizes precluded us from testing for population differentiation between the two sampling years in these sites.

Overall, within-site relatedness among coyotes was higher than between-site relatedness (Figure 3), with 58% of individual pairs within sites being first- or second-degree relatives. All sites had mean pairwise relatedness significantly larger than r = 0 (Figure S2). Between sites, only 11% of individual pairs were relatives. In the BFS, there was higher relatedness among coyotes within sampling years than between sampling years (Figure 3).

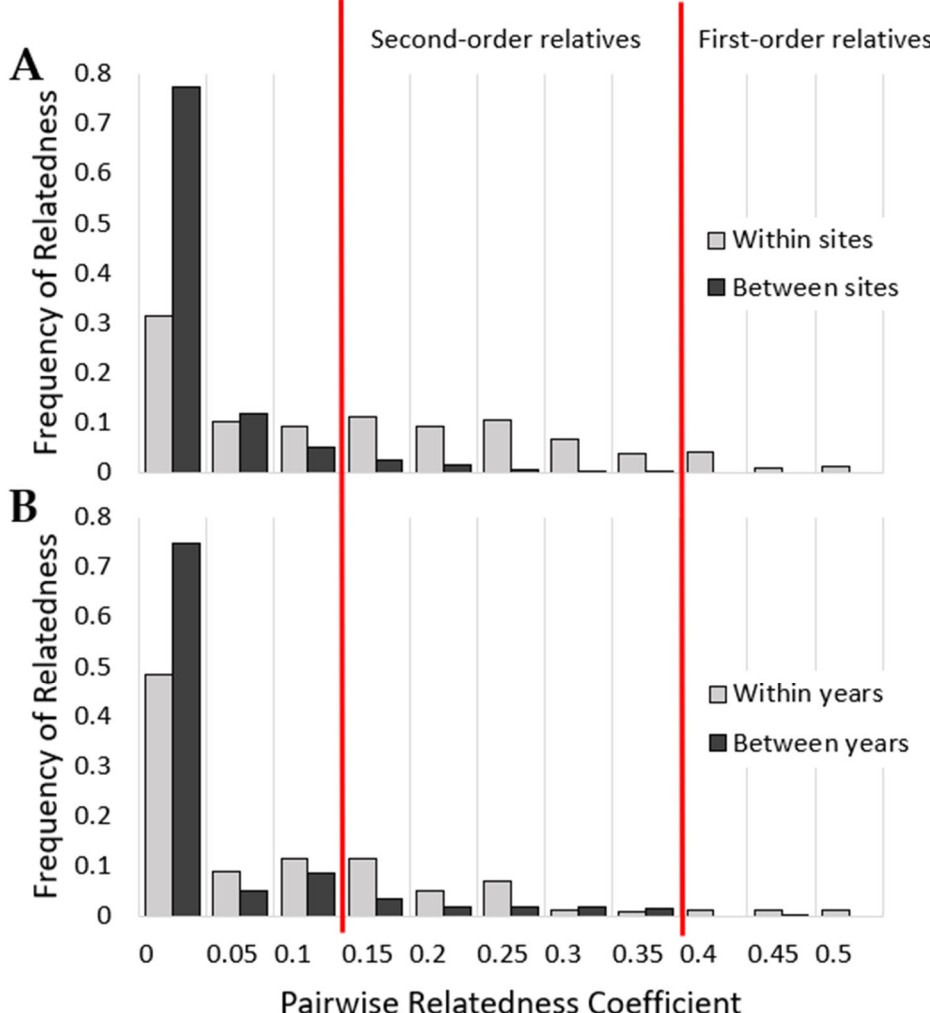

**Figure 3.** (**A**) Histogram of pairwise relatedness coefficients (Lynch–Ritland estimator) of coyotes within sites and between sites. (**B**) Histogram of pairwise relatedness coefficients (Lynch–Ritland estimator) of coyotes within sampling years and between sampling years in the BFS. In both plots, negative values were converted to zero. Red bars mark r = 0.1 and r = 0.4. Values between 0.1 and 0.4 indicate second-order relatives, and values greater than 0.4 indicate first-order relatives.

**Table 3.** Pairwise F$_{ST}$ values among four sites and *p*-values based on 9999 permutations.

|  | **BFS** | **CHWP** | **NEP** |
|---|---|---|---|
| CHWP | 0.112<br>*p* < 0.001 |  |  |
| NEP | 0.085<br>*p* < 0.001 | 0.128<br>*p* < 0.001 |  |
| CNP | 0.165<br>*p* < 0.001 | 0.180<br>*p* < 0.001 | 0.176<br>*p* < 0.001 |

## 4. Discussion

Our study emphasizes that incorporating a temporal component is critical to understanding the spatial population structure of coyotes. For example, while we did find coyotes in sites separated by a freeway to be genetically distinct, we found that all our coyote populations, even those in contiguous habitat, were distinct as well. This idea is further supported by a population structure of five genetic clusters, dividing coyotes among and within sampling sites. Examination of temporal changes in genetic structure and within-site relatedness indicates that coyotes may live and move in groups of closely related individuals. At the BFS, the fragment with the highest sample size, we found almost complete population turnover with only 4 of 29 individuals sampled in both years. The BFS coyotes formed two genetic clusters and had strong genetic differentiation between sampling years. The degree of differentiation between the two years at BFS (F$_{ST}$ = 0.050) is nearly the same as that between the BFS and NEP (F$_{ST}$ = 0.085). Furthermore, we found that, on average, coyotes sampled near in time were more closely related than those sampled further apart in time. The temporal patterns suggest the movement of family groups, likely breeding pairs and pack-associated offspring [38], in and out of our sites over time. Such movement would cause high fixation indices even in the absence of dispersal barriers among sites. Therefore, it is difficult to assess our hypothesis that the suburban matrix or freeways act as barriers to gene flow in southern California coyotes.

While many studies have examined the spatial population structure of coyotes in urban settings [20,21,39–41], none have considered the temporal structure of coyotes at specific sites. Many of these studies sampled coyotes over relatively large areas, opportunistically obtaining tissue from road-killed and harvested coyotes [40], or from tissue depositories [21,41]. This collection method allows for location to be treated as a continuous variable or for widely dispersed samples to be grouped into broad categories [21,40], thus allowing for different kinds of analyses than the present study. In contrast, our continuous sampling of the same sites over 15 months allowed us to observe a population turnover. While other studies had a longer sampling period—Riley et al. [20] sampled coyotes over 7 years, DeCandia et al. [41] included samples collected over 16 years, and Rashleigh et al. [39] and Damm et al. [40] sampled coyotes over 2 years—none examined temporal changes in population structure. If coyotes are indeed moving across the landscape in family groups, a sample of coyotes from the same area over many years may include different genetically structured groups. This may influence fixation indices, muddling our understanding of gene flow between coyotes sampled in different areas, and perhaps creating an illusion of gene flow barriers. In the present study, spatial structure alone suggested strong isolation of coyotes in our sites. However, an assessment of genetic variation and relatedness in one fragment across two years of sampling revealed two distinct groups and significant genetic differentiation between sample years—an indication that movement in and out of the site was not random but was done by groups of related individuals.

In addition to incorporating temporal structure, integrating animal tracking is critical to shed light on coyote movement patterns [42]. Riley et al. [20] accompanied genetic analyses of population structure with radio telemetry and found that coyotes on either side of a major freeway were genetically distinct despite the movement of individuals across the freeway. However, Riley et al. [20] admit that they did not track a sufficient number

of territorial individuals to assess territory overlap, suggesting that many of their tracked coyotes were transient. Therefore, they would not have been able to detect if individuals move in packs. Other tracking studies of individual coyotes in urban settings have observed movement across roads and among multiple habitat fragments, with territorial coyotes defending smaller home ranges and some individuals residing solely within small fragments of natural habitat [19,43,44]. As most tracking studies have focused on individual movements, it is difficult to ascertain how coyotes move in relationship to one another.

While coyotes can reside in packs both in natural and urban settings, generally consisting of breeding pairs and associated offspring acting as alloparents to younger siblings [45], they are more likely to share space and resources with each other in fragmented habitats than in more natural ones. For instance, highly related coyote family groups occupy small patches of greenspace in New York City [46]. Atwood and Weeks [47] found that 14 of 25 coyotes tracked in a fragmented landscape formed pairs (including non-breeding pairs) that interacted with each other spatially and temporally. Grubbs and Krausman [48] found individual coyotes from their tracked urban pack moving into new territories during their study period, but did not document a home-range shift of the entire pack. Other tracking studies conducted in developed areas documented home-range shifts of individual coyotes throughout the study periods, but not entire packs [49,50]. More information is needed to understand the nature of this movement through urban areas in light of the coyote's social structure. Future tracking studies should aim to capture and collar all pack members (coyotes residing within the same territory), and if possible, multiple packs residing in adjacent territories, in conjunction with incorporating molecular techniques to assess relatedness of tracked coyotes. Specifically, the recent advancement of SNP genotyping of non-invasive samples [51] may provide more detailed genetic data to identify individuals and assess relatedness and population structure. Combined tracking and molecular approaches focused on packs could provide a deeper understanding of how pack behavior and movement relates to overall population structure in a fragmented landscape.

The extent to which coyote mortality may explain the population turnover within sites, particularly urban habitat fragments, requires further examination. Anecdotally, high rates of sarcoptic mange, hair loss caused by the mite *Sarcoptes scabiei*, were reported for coyotes in the CWP during our study period (W.M.M. pers comm.). Coyotes with mange suffer from reduced body weight, reduced fat deposits, impaired thermoregulation, and increased energy demands, contributing to higher rates of mortality [52,53]. Urban coyotes are also highly vulnerable to mortality caused by vehicle collisions [43] and by consumption of anticoagulant rodenticides [49]. Generally, anthropogenic mortality in mammals increases with human impacts on the landscape [54]. As mortality clears out once-occupied territories, new coyotes will likely move in [55]. This cycle of territory-owner mortality and uptake of territory by new individuals and, possibly, new family groups, could be contributing to the temporal structure we observed.

Another possibility is that isolated fragments of natural habitat in this urban setting are not sufficient to support territorial coyotes for long periods of time. In contrast to previous studies, Riley et al. [49] found that in southern California there was a positive relationship between coyote home-range size and degree of urban association. Similarly, Gese et al. [56] found that coyotes in developed areas had home ranges twice the size of those in non-developed areas. In the Chicago metropolitan area, Zepeda et al. [57] found that the degree of surrounding development increased a coyote's dispersal probability and also their dispersal distance. This could indicate that urban areas are less suitable than natural areas [49], and that coyotes must expand their home range to meet their feeding, resting, and denning needs [56]. As space-sharing increases with habitat fragmentation [47], the small natural fragments in the present study might harbor an unsustainable density of coyotes for the resources they offer, causing packs to vacate the fragments over time to seek new territories. There is a threshold of size and isolation for natural habitat fragments to contain coyotes [58], but further research should investigate necessary attributes of fragments to sustain coyote packs long-term.

Though our study is limited by a small sample size (68 individuals genotyped at 7 microsatellite loci) and a relatively short temporal span (15 months), we were able to ascertain the spatial and temporal genetic structure of coyotes in four sites. Further studies would benefit from sustained sampling of habitat fragments over longer periods of time as well as increased sampling effort in surrounding habitat to possibly re-capture migrating coyotes or identify the source populations of coyotes using isolated fragments. Increased sampling efforts may help us better understand the forces driving coyote population structure in fragmented suburban landscapes by allowing us to disentangle natal dispersal, physical barriers, mortality, and transient habitat use.

## 5. Conclusions

Our study emphasizes that incorporating temporal components of genetic variation is critical to understanding genetic barriers in a fragmented landscape. Although the use of genetic tools to elucidate the effects of habitat fragmentation has suggested that urban environments represent significant barriers to gene flow, this contrasts with our general sense that coyotes move freely in these environments. Closer examination of temporal genetic structure within fragments suggests that coyotes are moving throughout the landscape in packs of closely related individuals, creating the illusion of high degrees of genetic differentiation among sites. Consequently, we are uncertain of how suburban development and freeways influence gene flow. Our research suggests that behavioral and genetic studies of urban coyote pack movement and pack social structure are critical to understanding how coyotes interface with fragmented landscapes. Future studies should combine GPS tracking of entire packs with molecular genetic techniques to examine both spatial and temporal dynamics. With more integrated research, land managers will be better equipped to ensure the preservation of this important apex predator in natural fragments of southern California.

**Supplementary Materials:** The following supporting information can be downloaded at: https://www.mdpi.com/article/10.3390/d15040498/s1, Table S1. Primer sequences for microsatellites and cytochrome-b region for species identification. Figure S1. Genetic assignments from STRUCTURE, showing different levels of K. Figure S2. Mean within-site pairwise relatedness.

**Author Contributions:** S.B., J.E.M. and W.M.M.III conceived and designed the research; S.B. collected the samples; J.D.M., J.E.M. and S.B. designed the laboratory methodology; S.B. and J.E.M. performed the laboratory work; S.B. led the analyses, with guidance from J.E.M., J.D.M. and W.M.M.III; S.B. led the manuscript writing and was assisted by J.E.M., J.D.M. and W.M.M.III; all authors revised and corrected the final versions of the manuscript. All authors have read and agreed to the published version of the manuscript.

**Funding:** This research was funded by the Pomona College Research Committee: The Pomona College Seaver Funds and the Biology Senior Thesis Funds.

**Institutional Review Board Statement:** The animal study protocol was approved by the Institutional Animal Care and Use Committee of Pomona College (Protocol #R/SMEYBIRRES-90817, 8 September 2017).

**Data Availability Statement:** The data presented in this study are openly available in Zenodo at DOI 10.5281/zenodo.7697792, https://zenodo.org/record/7697792 accessed on 3 March 2023.

**Acknowledgments:** We thank the Pomona College Research Committee for an award supporting this work. We thank the Bernard Field Station, California Botanic Garden (specifically Fran Lehman), Claremont Hills Wilderness Park, Chaffey Nature Preserve of Chaffey College (specifically Sarah Cotton), and the North Etiwanda Preserve (specifically Erin Opliger) for allowing us to collect scat on their premises. We also thank Nelson Ting for allowing us to complete laboratory work within his lab space at the University of Oregon. Finally, we thank the anonymous reviewers for their role in improving the first draft of this manuscript.

**Conflicts of Interest:** The authors declare no conflict of interest.

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
