# Peer review of "An Illusion of Barriers to Gene Flow in Suburban Coyotes (Canis latrans): Spatial and Temporal Population Structure across a Fragmented Landscape in Southern California"

_diversity, doi:10.3390/d15040498_

Round 1
Reviewer 1 Report
This well written manuscript investigates some interesting questions about spatial vs temporal population structure in peri-urban coyotes. It was a pleasure to read about this study! I think the authors could improve the manuscript by discussing some of the limitations of their study, including the relatively short temporal span (14-15 months) and small sample sizes (5-7 loci for 68 unique individuals). I also have some more specific comments below.
General comments
(1) Given the small number of genetic loci which had only moderate allelic richness, how confident are the authors that identical genotypes = same individual. Could individuals share identical genotypes? This might be a limitation of the small number of microsatellite markers employed.
(2) Looking at Figure 2, did individuals from the same cluster (yellow, orange, red, blue or green) share higher relatedness? Perhaps I missed this being discussed. IE is the population structure observed related to pack or family groups moving transiently through the landscape - particularly BFS.
(3) At Line 209 the text states “All of our sites showed a population turnover throughout the study period” however looking at Figure 2, CNP does not appear to have undergone a population turnover. CNP also has lower total Ho, He and Ar values. NEP also appears to have had limited turnover. The authors could consider running a disperser analysis in Geneclass2, there does look to be a possible dispersal from BFS to NEP.
(4) When running STRUCTURE it is general practice to run 10 replicates per K value and average across those 10 replicates. I would suggest that the authors run some additional Structure modelling to increase their replicates from 5 to 10. Additionally given K=3 and K=5 both have peaked delta K values it may be worth discussing both scenarios in the results. Additionally the authors should consider running Structure with the LocPrior option which often performs well for datasets with weak population structure. The correlated allele frequency model may also be more appropriate to the dataset as it does not assume knowledge about the allele frequencies in populations so is more conservative. The authors may find the discussion of Structure run parameters in this paper is helpful: https://doi.org/10.3389/fgene.2013.00098.
Specific / minor comments
Figure 1: The scale bar is in miles but the distances given are metric, this is a little bit confusing.
L166: why were only 5 loci used for individual and sibling id when there were 7 genotyped?
L198: how many samples were excluded as they carried identical genotypes?
Table 2: the authors should report standard error for Ho, He and Ar.
L 17 vs 219: In the abstract it reads as if a single site had turnover; “near-complete population turnover and examination of temporal genetic structure and relatedness within one site”, in the results you state that all the sites had population turnover, Figure 2 shows turnover in 2-3 of the populations but little evidence of turnover in the CNP population.
L316: replace multiple years with 15 months.
L357: use of SNP genotyping ie RAD-seq appropriate to non-invasive samples may also provide more detailed genetic data to identify individuals, relatedness and population structure in future studies.
L362: Investigating temporal population structure over a longer period of time may help identify the circumstances driving coyote population structure in this region ie natal dispersal, genetic barriers, mortality or transient habitat use. Additionally, gathering additional genetic data from other sites to better characterise the genetic identity of coyotes in the region would probably help identify population structure patterns in future research.
L371: where there any coyote culls carried out in the region?
L377-393: Collecting more genetic data from coyotes in the surrounding area may identify the source populations of coyotes using the isolated fragments, really interesting!
Author Response
Please see the attachment. Thank you very much for your thoughtful remarks and suggestions!

Reviewer 2 Report
The authors investigate how urban structures may represent barriers that may impede gene flow in city-dwelling animals such as coyotes. This topic is especially interesting because it has been addressed taking into account population dynamics both in space and in time.
While the authors do not reach definitive conclusions about how suburban developments and highways affect coyote gene flow in urban environments, their findings demonstrate the importance of genetic and behavioral studies to understanding how coyotes interact with landscapes fragmented.
This is a well done paper with an appropriate methodology. I have no particular suggestions to make other than, if possible, to reduce the discussion by shortening the various hypotheses that are merely speculative.
Author Response
Please see the attachment. Thank you very much for taking the time to review this article and providing your feedback!
